

# Organic acids and 2,4-Di-*tert*-butylphenol: major compounds of *Weissella confusa* WM36 cell-free supernatant against growth, survival and virulence of *Salmonella* Typhi

Wattana Pelyuntha[1], Chaiyavat Chaiyasut[1], Duangporn Kantachote[2] and Sasithorn Sirilun[1]

[1] Innovation Center for Holistic Health, Nutraceuticals and Cosmeceuticals, Faculty of Pharmacy, Chiang Mai University, Chiang Mai, Thailand
[2] Department of Microbiology, Faculty of Science, Prince of Songkla University, Hat Yai, Thailand

## ABSTRACT

**Background:** *Salmonella* Typhi (*S*. Typhi), the causative agent of typhoid fever, causes serious systemic disease in humans. Antibiotic treatment is required for the *S*. Typhi infection, while the inappropriate use of antibiotics causes increased drug-resistant *S*. Typhi. Hence, alternative therapies through non-antibiotic approaches are urgently needed. The use of beneficial lactic acid bacterium and/or its metabolites to control typhoid fever represent a promising approach, as it may exert protective actions through various mechanisms.

**Method:** In this study, the cell-free culture supernatant (CFCS) of *Weissella confusa* WM36 was evaluated via the antibacterial activity, and its metabolites were identified. In addition, the effects of CFCS on *Salmonella* virulence behaviors were also investigated.

**Result:** Based on strong inhibition the growth of *S*. Typhi DMST 22842, organic acids (lactic acid and acetic acid) and 2,4-Di-*tert*-butylphenol (2,4 DTBP), were the main antibacterial metabolites presented in CFCS of strain WM36. Minimum inhibitory concentration (MIC) at 40% WM36–CFCS dramatically reduced the *S*. Typhi population to more than 99.99% at 4 h and completely inhibited biofilm formation, while sub-MIC at 20% (v/v) and MIC could reduce 100% of motility. Additionally, sub-MIC at only 10% (v/v) WM36–CFCS did down-regulate the expression of virulence genes which are responsible for the type-III secretion system, effector proteins, and quorum sensing system in this pathogen.

**Conclusion:** *W. confusa* WM36 and its metabolites are shown to be a promising candidates, and an effective approach against typhoid *Salmonella* burden.

## INTRODUCTION

Typhoid fever is a serious bacterial infection caused by *Salmonella enterica* serotype Typhi (*S*. Typhi). This bacterium is transmitted through the consumption of contaminated

Corresponding author
Sasithorn Sirilun,
sasithorn.s@cmu.ac.th

foods and water (direct fecal-oral transmission) and infects only humans (*De Jong et al., 2012*). Worldwide, typhoid fever was estimated to have 12 million cases and 130,000 deaths in the year 2010 (*John, Van Aart & Grassly, 2016*). It is a systemic disease of varying severity, with symptoms including a gradual onset of high fever, headaches, weakness, dry cough, abdominal pain or constipation. The severity of infections in humans depends on the health status of the human host. Young children, adults, and immunocompetent patients are more susceptible than healthy adults (*Eng et al., 2015*). *S.* Typhi invades the small intestine, enters the blood circulation, and survives within the phagocytic cells. This bacterium also colonizes the spleen, liver and bone marrow in infected humans. In addition, it invades the defense mechanisms to interrupt the host immune responses (*Raffatellu et al., 2008*).

*S.* Typhi pathogenesis relies mainly on the expression of genes within *Salmonella* pathogenicity islands (SPI); all islands have their own patterns of regulation and expression of their genes. SPI-1 is one of them and encoded for type-III secretion system (T3SS). The T3SS is a syringe-like molecule, that delivers bacterial effector proteins into the host cell to rearrange cytoskeleton to aid bacterial invasion, survival and multiplication in host cells (*Kaur & Jain, 2012*; *Marcus et al., 2000*). Moreover, *S.* Typhi consists of other virulence factors including fimbriae, flagella, and biofilms, which are responsible for adherence, motility, and resistance to antibiotics, and bactericidal agents, respectively (*De Jong et al., 2012*). Additionally, quorum sensing (QS) systems have been described in *Salmonella* virulence, these systems work as a signaling network to control the expression of behaviors, especially virulence gene expression. SPI-1 genes and biofilm of *Salmonella* are believed to be controlled by QS systems (*Pui et al., 2011*).

Up-to-date antibiotic therapy is obligately required for the treatment of typhoid fever. The first line antibiotics suggested in healing illness including chloramphenicol, ampicillin, and trimethoprim-sulfamethoxazole (*Mutai et al., 2018*). Increased inappropriate use of first line drugs has resulted in the emergence of multidrug-resistant (MDR) *S.* Typhi. Fluoroquinolones and the third-generation drugs have become alternative treatments of typhoid fever, but these drugs can cause undesirable side effects (*Veeraraghavan et al., 2018*). The Center for Disease Control and Prevention ranks the incidence of typhoid fever that emerges by MDR-*S.* Typhi spread as a serious threat. Monitoring and preventing programs are required to reduce the spread of this resistant strain (*Mutai et al., 2018*). In this regard, the use of alternative therapies for controlling bacterial infections caused by antibiotic-resistant bacteria through the non-antibiotic approach is urgently needed for treatments.

Lactic acid bacteria (LAB) are presenting as a promising candidate, as it may exert protective actions through various mechanisms. LAB can modulate the host's immune system, compete for adhesion sites of pathogens on epithelial cells, and produce a variety of antimicrobials against many pathogenic microorganisms (*Both et al., 2011*). It has long been known that most LAB strains are recognized as safe (GRAS), and several strains used in food manufacturing and nutraceutical developments. They are normally found as a dominant normal flora in the gastrointestinal tracts of mammals, as well as in several types of fermented foods. LAB can utilize the sugars as a sole carbon source for energy and

change into most organic acids as the natural preservatives in fermented foods. Hence, they are considered to be safe, and act as a multifunctional agent in food production (*Pandey, Naik & Vakil, 2015*). The antimicrobial substances produced by LAB are divided into two groups based on molecular weight, low molecular mass substances (<1,000 Da) including organic acids, $H_2O_2$, acetoin, diacetyl, acetaldehyde, etc., and high molecular mass substances (>1,000 Da) such as bacteriocins (*Šušković et al., 2010*).

The lactic acid bacterial genus *Weissella* was proposed as a genus re-classified *Lactobacillus* and *Leuconostoc* spp. This genus is frequently detected in several fermented foods (*Fessard & Remize, 2017*). *Weissella confusa* has been established from a variety of foods, such as fermented vegetables and fruits; milk and milk products; and acid-rich carbohydrate foods. However, it can also be isolated from the sewage and clinical samples, and this strain is believed to be normal microflora in human gastrointestinal and urogenital tracts (*Fusco et al., 2015*). The antagonistic activities of *W. confusa* on pathogenic bacteria including *Bacillus cereus*, *Escherichia coli*, *Helicobacter pylori*, *Proteus mirabilis*, *Pseudomonas aeruginosa*, and *Staphylococcus aureus*, have been described in several previous reports. *W. confusa* can produce organic acids and bacteriocins in culture medium and some strain can strongly adhere to intestinal epithelial cells (*Chavasirikunton, Vatanyoopaisarn & Phalakornkule, 2006*; *Lee, 2004*; *Nam et al., 2002*; *Purkhayastha et al., 2017*; *Shah et al., 2016*). An increasing number of LAB and metabolite related, studies only focus on the properties of other LAB rather than *Weissella* spp. isolates. *W. confusa* was therefore chosen in this study, and its antagonistic activities were also examined to provide more information on LAB.

Organic acids are commonly characterized as antimicrobial substances produced by all LAB strains, while their amounts and types depend on LAB strains and activity in a culture medium. Organic acids are effective at inhibiting the growth of Gram-positive and Gram-negative bacteria, which belong to yeasts and molds (*Pessione, 2012*; *Rattanachaikunsopon & Phumkhachorn, 2010*). Diacetyl, acetaldehyde, and acetoin are characterized as antimicrobial substances produced by a variety of heterofermentative LAB. These compounds are not the major antimicrobial substances, but they work together, along with other antimicrobial substances. $H_2O_2$, $CO_2$, fatty acids, and cyclic dipeptides produced by LAB act as microbicide against Gram-negative and Gram-positive bacteria, yeasts, and molds (*Dicks et al., 2018*; *Pessione, 2012*; *Rattanachaikunsopon & Phumkhachorn, 2010*). Interestingly, one of the most identified antimicrobial substances produced by LAB is bacteriocins, which are secondary metabolites. Their main function is to inhibit or kill undesired microorganisms in food products and are safe food biopreservatives obviating the need for harmful chemical preservations (*Dicks et al., 2018*; *Pessione, 2012*; *Rattanachaikunsopon & Phumkhachorn, 2010*; *Silva, Silva & Ribeiro, 2018*). Moreover, LAB and their metabolites are not only used to extend the shelf-life of foods from spoilage microorganisms, but are also effective in the prevention and treatment of bacterial infections. There is ample diversity of literature supporting the efficacy of LAB and their metabolites on pathogenic microorganisms. Therefore, potent LAB strains provide a better alternative way to fight bacterial infections.

The objectives of this study are (i) to extract and evaluate the antagonistic activity of cell-free culture supernatant (CFCS) of *W. confusa* WM36 against the causative agents of typhoid fever, *S.* Typhi, (ii) to investigate the effect of WM36–CFCS on *Salmonella* growth, biofilm formation, motility and virulence gene expression in vitro, and (iii) to characterize the compounds of antimicrobial substances produced by WM36 strain using suitable methods via High-Performance Liquid Chromatography (HPLC) and Gas Chromatography-Mass Spectrometry (GC-MS).

## MATERIALS AND METHODS

### Bacterial strain and growth conditions

*Weissella confusa* WM36 isolated from fermented grape was obtained from the Innovation Center for Holistic Health, Nutraceuticals and Cosmeceuticals, Faculty of Pharmacy, Chiang Mai University. *W. confusa* WM36 is permanently deposited in the Thailand Bioresource Research Center (TBRC), Pathum Thani, Thailand with the Accession Numbers TBRC11086. The WM36 was cultivated in de Man Rogosa and Sharpe (MRS) medium at 37 °C overnight prior to analysis. *Salmonella* Typhi DMST 22842 used in this study was obtained from the Department of Pharmaceutical Science, Faculty of Pharmacy, Chiang Mai University, and was cultivated at 37 °C in Luria-Bertani (LB) medium or tryptic soy broth (TSB).

### Preparation of *W. confusa* WM36–CFCS

To obtain CFCS, *W. confusa* WM36 was activated and cultured in the MRS broth at 37 °C for 24 h with the final concentration number at $10^6$ CFU/ml. CFCS was collected by centrifugation at 6,000 rpm at 4 °C for 15 min and sterilized through a 0.22 μm syringe filter and kept in −20 °C.

### Antibacterial susceptibility test of CFCS

The modified agar diffusion method described by Reis et al. (2016) was provided. *S.* Typhi was pre-cultured in TSB at 37 °C for 18 h. Then, the pathogen was diluted with 0.85% normal saline solution (NSS) and yielded the final concentration at $10^5$ CFU/ml in TSA soft agar (1.0% agar). A 20 ml of soft agar mixture was poured into a sterile petri-dish containing stainless-steel carriers (five mm in diameter). After setting of the agar medium, wells were formed by pulling out the carriers; 50 μl of CFCS was dropped into each well (three wells per plate). All plates were incubated at 37 °C for 48 h. The inhibitory zone in the vicinity of the well was observed and the diameter was measured with the Vernier caliper in millimeter measurement. Five mg/ml of ampicillin and fresh MRS broth served as the positive and negative control for this study.

For minimum inhibitory concentration (MIC) test, the broth microdilution method was performed to determine the MIC value of CFCS. The sterile CFCS was serially diluted in 96-well round-bottom microtiter plates containing LB broth to yield the final concentrations ranging from 5% to 90% of CFCS. The inoculum of *S.* Typhi (10 μl) containing ~$10^5$ CFU/ml was added to each well. The fresh LB broth well was reserved as the sterility control (no inoculum), and the inoculum viability (no CFCS) was served

as a positive control. All microplates were incubated at 37 °C for 24 h. MIC values were defined as the lowest concentration of CFCS that had no visible growth (*Ben Slama et al., 2013*).

## Characterization of antimicrobial substances in CFCS

To eliminate the antimicrobial effect of organic acids, WM36–CFCS was neutralized as pH to 7.0 with 1 N NaOH to rule out acid inhibition, proteinase K and lysozyme to rule out proteinaceous substances inhibition, and catalase to rule out $H_2O_2$ inhibition. An antagonistic activity of these neutralizing CFCS was investigated using the agar well diffusion method as previously described. Untreated CFCS served as the control (*Therdtatha et al., 2016*).

## pH, titratable acidity, and organic acid analysis

The pH value of WM36–CFCS was measured with a pH meter. For the titratable acidity (TA) assay, one ml of WM36–CFCS was diluted with 9 ml of deionized water in 250 ml of Erlenmeyer flask. Two drops of phenolphthalein solution were added as indicator. The mixture was titrated with standardized NaOH (∼0.1 N) until a faint lasting pink color appeared and the titratable volume was recorded. TA was expressed as the percentage of lactic acid equivalent (% LAE):

$$(\% \text{ LAE}) = [(\text{Normality of standardized NaOH} \times \text{Titratable volume of standardized NaOH} \times 90)/\text{sample volume} \times 1000] \times 100.$$

Lactic acid and acetic acid in WM36–CFCS were investigated by HPLC with the slightly modified method described by *Lin et al. (2011)*. WM36–CFCS was diluted ten-fold with deionized water and filtered through 0.22 μm syringe filter and kept in an autosampler vial. Aliquot of 20 μl was injected into a 250 × 4.6 mm ACE Generix5 C18 column (ACE®, Westminster, UK). Elution was performed at 30 °C with 0.1% phosphoric acid at a flow rate of 0.5 ml/min. The optical density of two organic acids was measured at 210 nm with an SPD-20 UV detector (Shimadzu, Kyoto, Japan). To quantify organic acids, the different organic concentrations, which ranged from 1 to 30 mm were used as standards.

## Gas chromatography-mass spectrophotometry

The metabolites within the CFCS were analyzed using Agilent 7890A Gas chromatograph (Agilent Technology, Santa Clara, CA, USA) interfaced to Agilent 5975C (EI) mass-selective detector. The mass spectra were scanned in the mass range of 43–550 u at a rate of 0.99 scans/s. Injector, interface and ion source temperatures were held at 260, 300 and 230 °C, respectively. A DB-5MS column (30 m × 0.25 mm I.D., 0.25 μm film thickness, Agilent Technology, Santa Clara, CA, USA) was used for all analyses. Helium was used as the carrier gas at a constant flow rate of 0.5 ml/min for all analyses. The oven temperature was set initially at 100 °C (2 min), then increased to 250 °C at a rate of 5 °C/min and finally programed to 300 °C at a rate of 20 °C/min (5 min) (modified method from *Lee et al., 2012*).

## Influence of CFCS on *Salmonella* growth and survival

*Salmonella* Typhi were pre-cultured in a suitable medium and conditions as previously described. After that, this bacterium was adjusted to obtain the final concentration of approximately at $10^6$ CFU/ml in LB broth. Then, *S.* Typhi was transferred and cultured in the same medium supplemented with CFCS of WM36 at MIC values in 100 ml Erlenmeyer flask and incubated at 37 °C. The culture broth was sampled at 0, 2, 4, 8, 12, 16, 20 and 24 h of incubation. Then each of the samples were serially 10-fold diluted with 0.85% NSS to obtain the appropriated dilutions. A 0.1 ml of diluted tubes was spread on LB agar (1.5% agar) plates for *Salmonella* count. The culture of *S.* Typhi without CFCS served as the control group. All plates were incubated at 37 °C for 24–48 h (*Shah et al., 2016*). The number of viable cells were counted and calculated as the percentage of *Salmonella* survival with a given formula;

$$\% \; Salmonella \; survival = (Viable \; counts \; in \; treatment / Viable \; counts \; in \; control) \times 100$$

## Influence of CFCS on *Salmonella* motility

*Salmonella* Typhi was pre-cultured in LB broth at 37 °C overnight. Two microliters of the culture were dropped on the center of the LB agar (0.5% agar) containing the MIC and sub-MICs of CFCS and incubated at 37 °C for 24 h. Diameter zones of motility were measured (*Choi et al., 2012*).

## Influence of CFCS on *Salmonella* biofilm formation

The effect of CFCS against *Salmonella* biofilm formation was tested on 24-well polystyrene microtiter plates containing a sterile piece of glass (15 × 8 mm). CFCS at MIC and sub-MICs were added in LB broth containing the bacterial suspension at $10^6$ CFU/ml. The plates were incubated statically at 37 °C for 48 h. After incubation, the glass piece was stained with 0.4% crystal violet solution, washed three times with sterile distilled water and air-dried at room temperature under biosafety cabinet Class 2. Stained glass was placed on slides with the biofilm in an upright position, and investigated by stereomicroscope (Stemi 508; Zeiss, Oberkochen, Germany) at magnifications of X10. Visible biofilms were captured with an attached digital camera (Axiocam 105 color; Zeiss, Oberkochen, Germany) and analyzed through the ZEN 2 software program. Negative control wells which contain only LB were added in each assay, while suspension of *S.* Typhi in LB broth was included as a positive control to determine the bacterial adherence and biofilm formation without CFCS (*Jeong et al., 2018*; *Kang et al., 2006*; *Taheur et al., 2016*).

## Influence of CFCS on *Salmonella* virulence gene expression

*S.* Typhi was cultured in LB broth supplemented with CFCS at MIC and sub-MICs. Bacterial cells were harvested and washed three times with NSS by centrifugation at 6,000 rpm, for 15 min at 4 °C. Bacterial total RNA was extracted using Trizol® reagent (Invitrogen, Waltham, MA, USA) according to the manufacturer's instructions with slight modification. Briefly, one ml of Trizol® reagent was added and incubated for 5 min to dissociate of nucleoprotein complexes, followed by 0.2 ml of chloroform and incubated for

3 min. The mixture was centrifuged for 15 min at 12,000X$g$ at 4 °C. An aqueous phase containing RNA was transferred to a new tube and reacted with 0.5 ml of isopropanol, followed by incubating at 4 °C overnight. After incubation, the mixture was centrifuged at 12,000X$g$, for 10 min at 4 °C. After that, the supernatant was discarded by carefully pouring; the white gel forming pellets containing RNA appeared at the bottom of the tube. A pellet was suspended with 0.5 ml of 75% ethanol and centrifuged at 7,500X$g$, for 5 min at 4 °C. The supernatant was discarded; RNA was air-dried for 15 min, suspended in 50 μl with nuclease-free water, heated with heat block at 60 °C for 15 min and kept at −20 °C.

After RNA extraction, the total RNA was quantified by Qubit® RNA BR assay kit (Life Technologies, Waltham, MA, USA) with Qubit® 3.0 Fluorometer following the manufacturer's instruction. Then, cDNA was synthesized with a High-Capacity cDNA Reverse Transcription kit (Applied Biosystem, Waltham, MA, USA) according to the manufacturer's instruction. Briefly, 20 μl of reaction consists of 10 μl of 0.5 μg total RNA, 2 μl of 10X RT buffer, 0.8 μl of 25X dNTP, 2 μl of 10X random primer, 1 μl of multiScribe™ reverse transcriptase, 1 μl of RNase inhibitor and 3.2 μl of nuclease-free water. The thermal cycler was programed as 25 °C for 10 min, 37 °C for 120 min, 85 °C for 5 min, and 4 °C until the program was stopped. All cDNA reaction tubes were kept at −20 °C.

The qPCR assay was performed using a QuantStudio 6 Flex Real-Time PCR cycler (Thermo Fisher Scientific, Waltham, MA, USA) with the flowing thermal cycle profile: 50 °C for 2 min and 95 °C for 2 min for initial UDG activation; 40 cycles of 95 °C for 20 s (denature) and 60 °C for 20 s (annealing/extension). A melting curve was generated by heating 95 °C for 20 s, 60 °C for 20 s and ramping back to 95 °C in 0.05 °C increments. The 20 μl of reaction mixture contained 2 μl of cDNA samples, 10 μl of PowerUp™ SYBR® Green master mix (Applied Biosystem™, Waltham, MA, USA), 0.2 μl of each forward and reverse primer (Table 1), and 7.6 μl of nuclease-free water. The levels of virulence gene expression were normalized using 16s rDNA as an internal housekeeping gene, and fold change of target genes was calculated by the ΔΔCt method. All values derived from $2^{-\Delta\Delta Ct}$ represent fold changes of samples in abundance relative to the reference sample. The reference samples had the $2^{-\Delta\Delta Ct}$ value of one. The treatment not subjected to CFCS treatment served as the reference sample (*Bayoumi & Griffiths, 2010*; *Choi et al., 2012*; *Muyyarikkandy & Amalaradjou, 2017*; *Yang et al., 2014*).

### Statistical analysis

Statistical analysis was performed using SPSS (version 17.0) of Windows statistics software (SPSS Inc., Chicago, IL, USA). The data were subjected to analysis of variance followed by Tukey's range test. A difference was considered statistically significant at a *p*-value of less than 0.05.

## RESULTS

### Antibacterial susceptibility tests of CFCS

To evaluate the antagonistic activity of CFCS of *W. confusa* WM36 on *S.* Typhi, the antibacterial susceptibility test through agar well diffusion and the MIC test were performed. The CFCS of *W. confusa* WM36 showed an inhibitory effect against

**Table 1** Lists of forward and reverse primers used in this study.

| Genes | Primers (5′–3′) | Amplicon size (bp) | References |
|---|---|---|---|
| Housekeeping gene | | | |
| *16s rRNA* | Forward: CAGAAGAAGCACCGGCTAACTC | 87 | *Yang et al. (2014)* |
| | Reverse: GCGCTTTACGCCCAGTAATT | | |
| Virulence genes | | | |
| *hilA* | Forward: CATGGCTGGTCAGTTGGAG | 150 | *Yang et al. (2014)* |
| | Reverse: CGTAATTCATCGCCTAAACG | | |
| *hilD* | Forward: ACTCGAGATACCGACGCAAC | 129 | *Yang et al. (2014)* |
| | Reverse: CTTCTGGCAGGAAAGTCAGG | | |
| *sopB* | Forward: AACCGTTCTGGGTAAACAAGAC | 77 | *Yang et al. (2014)* |
| | Reverse: GGTCCGCTTTAACTTTGGCTAAC | | |
| *sopE2* | Forward: GCCTGCATCAACAAACAGACA | 72 | *Yang et al. (2014)* |
| | Reverse: ATACCGCCCTACCCTCAGAAG | | |
| *sipA* | Forward: GGCTTGCGTGCGGAAATA | 69 | *Yang et al. (2014)* |
| | Reverse: ATCGCTACATTGCGCTTTCA | | |
| *sipC* | Forward: CTGTGGCTTTCAGTGGTCAG | 150 | *Yang et al. (2014)* |
| | Reverse: TGCGTTGTCCGGTAGTATTTC | | |
| *sptP* | Forward: ATGCTCGTGCCTGGTGGTGTTA | 236 | *Yang et al. (2014)* |
| | Reverse: ACGGTAACGGCTGGTGATCT | | |
| *invF* | Forward: GCAGGATTAGTGGACACGAC | 87 | *Choi et al. (2012)* |
| | Reverse: TTTACGATCTTGCCAAATAGCG | | |
| *sdiA* | Forward: AGCAGTTTACGCTGCTCCTC | 164 | This study |
| | Reverse: GCCGTCCACTTCAGAATCTC | | |
| *luxS* | Forward: AGCATCTGTTTGCTGGCTTT | 178 | This study |
| | Reverse: TCCTGCACTTTCAGCACATC | | |

representative *S.* Typhi with the average inhibitory zone (including five mm diameter of well) as 8.17 ± 0.29 mm using the agar well diffusion method. The average CFCS inhibitory zone is lower than the positive control (five mg/ml ampicillin) that yielded an average activity of 12.75 ± 0.75 mm. The MIC value is 40.00% (v/v) and completely inhibited the visible growth in tested wells.

## Characterization of antimicrobial substances in CFCS

From the results, acid-neutralizing CFCS showed no inhibitory clear zone in the vicinity of observed wells when compared to protein-neutralizing, catalase-neutralizing, and untreated CFCS (Fig. S1). This indicated that the organic acids are the main antimicrobial substances found in WM36–CFCS. However, WM36–CFCS may have other antimicrobial metabolites in addition to organic acids, which require further investigation.

## pH, titratable acidity and organic acid analysis using the HPLC technique

The values of pH and TA (% lactic acid equivalent; LAE) of CFCS are shown in Table 2; Table S1. The property of WM36–CFCS was low pH. This property is assumed to be the

**Table 2 pH, titratable acidity (% LAE) and organic acids in WM36–CFCS.** All values provided as mean ± standard deviations of triplicate (*) or duplicate (**).

| pH* | % LAE* | Organic acids (mM)** | |
| --- | --- | --- | --- |
| | | Lactic acid | Acetic acid |
| 4.53 ± 0.01 | 1.10 ± 0.05 | 266.70 ± 0.81 | 261.33 ± 7.92 |

main anti-salmonella substances are acids. *W. confusa* WM36 can produce lactic acid and acetic acid up to 266 mm (24.02 g/l) and 261 mm (15.96 g/l), respectively in CFCS (Table 2). The chromatogram of organic acids in standard solution and the WM36 sample are shown in Fig. S2.

## Identification of metabolites in CFCS by GC-MS analysis

WM36 metabolites were identified and are listed in Table 3. The results collected from GC-MS analysis (Table 3; Dataset S1) revealed that 2,4 DTBP (No. 1) showed the most abundance. This metabolite showed the retention time at 13.450 min with the highest % peak area of 59.24. In addition, the other metabolites are also represented in this CFCS, but they showed lower abundance than No.1 compound.

## Influence of CFCS on *Salmonella* growth and survival

To study the effect of CFCS on *Salmonella* growth and survival, the MIC of WM36–CFCS was chosen as the most effective inhibition concentration. As shown in Table 4, there was no significant difference in the *Salmonella* population between the control and treatment at the starting time ($t = 0$). Significant differences were observed at longer incubation times (at 2–24 h) due to the effect of WM36–CFCS on the growth of *S.* Typhi in the treatment (Table 4; Table S2). *Salmonella* in the control significantly increased from $1.56 \times 10^6$ to $8.95 \times 10^7$ CFU/ml in 4 h until the end of cultivation at 24 h ($3.12 \times 10^8$ CFU/ml). In contrast, the treatment of *S.* Typhi with CFCS at MIC showed a remarkable decrease of this pathogen from $2.07 \times 10^6$ to $5.21 \times 10^5$ and $3.47 \times 10^3$ CFU/ml in 2–20 h of incubation. The results between the control and treatment indicate that CFCS at MIC significantly decreased the pathogen by starting within 2 h of the incubation test. In the period of time from 4 to 20 h, the MIC of CFCS caused <1% survival in the pathogenic population compared with the starting time (0 h).

## Influence of CFCS on *Salmonella* motility

Table 5 and Table S3 show the swarming zone of *S.* Typhi in the presence of different concentrations of CFCS at sub-MICs and MIC values. The swarming motility of *S.* Typhi at 10% (v/v) (Fig. 1B ) concentration gave similar results to the control (Fig. 1A) with no significant differences ($p > 0.05$). At 20% (v/v) (Fig. 1C) and MIC (Fig. 1D), were very effective concentrations, which completely reduced the swarming motility of *S.* Typhi; and there were significant differences ($p < 0.05$) compared with the control.

**Table 3 Metabolites in cell-free culture supernatant of *W. confusa* WM36.**

| Number | Chemical constituents | Area (%) | Retention time (min) |
|---|---|---|---|
| 1 | 2,4-Di-*tert*-butylphenol | 59.24 | 13.450 |
| 2 | Unknown | 1.89 | 16.107 |
| 3 | Unknown | 0.44 | 17.706 |
| 4 | 1, 3, 6, 10-Dodecatetraene | 1.26 | 18.117 |
| 5 | 3, 5-Di-*tert*-butyl-4-hydroxybenzaldehyde | 0.63 | 19.010 |
| 6 | Myristic acid | 0.34 | 19.126 |
| 7 | Unknown | 0.65 | 19.305 |
| 8 | 2-Propoxyethyl 4-(2, 4-dimethoxyphenyl)-2-methyl-5-oxo-1, 4, 5, 6, 7, 8-hexahydro-3-quinolincarboxylate | 2.75 | 19.408 |
| 9 | Unknown | 1.05 | 20.080 |
| 10 | Unknown | 0.47 | 20.890 |
| 11 | Phthalic acid | 0.62 | 21.124 |
| 12 | Cyclohexadecane | 0.50 | 21.577 |
| 13 | Unknown | 0.93 | 22.030 |
| 14 | Unknown | 0.90 | 22.359 |
| 15 | Methyl hexadecanoate | 0.59 | 22.448 |
| 16 | Methyl-3-(3,5-ditertbuthyl-4-hydroxyphenyl) propionic acid | 1.92 | 22.503 |
| 17 | Dibutyl benzene-1,2-dicarboxylic acid | 3.65 | 23.018 |
| 18 | Hexadecanoic acid | 0.37 | 23.196 |
| 19 | Ethyl 9-hexaadecenoate | 0.27 | 23.347 |
| 20 | Unknown | 0.43 | 23.807 |
| 21 | Unknown | 0.29 | 24.342 |
| 22 | Unknown | 0.22 | 25.145 |
| 23 | *p, p*'-diphenylmethane diisocyanate | 0.56 | 25.825 |
| 24 | Oleic acid, ethyl ester | 0.78 | 26.971 |
| 25 | 9-Octadecenoic acid (z)-, ethyl ester | 0.48 | 27.088 |
| 26 | Tetracosane | 0.28 | 30.945 |
| 27 | Pentacosane | 0.74 | 32.482 |
| 28 | 1, 2-Benzenedicarboxylic acid, mono (2-ethylhexyl) ester | 0.36 | 32.832 |
| 29 | Docosane | 1.50 | 33.553 |
| 30 | Heptacosane | 1.88 | 34.342 |
| 31 | Eicosane | 1.75 | 35.029 |
| 32 | Unknown | 0.27 | 35.111 |
| 33 | Nonacosane | 1.66 | 35.749 |
| 34 | Unknown | 0.53 | 36.051 |
| 35 | Unknown | 0.18 | 36.182 |
| 36 | Unknown | 1.37 | 36.539 |
| 37 | Unknown | 1.21 | 37.424 |
| 38 | Unknown | 0.99 | 38.467 |

**Table 4 Effect of WM36–CFCS at MIC (40%) on viable cells of *Salmonella* Typhi.** All values provide as mean ± standard deviation of triplicate, the asterisk (*) indicates the significant difference ($p < 0.05$) between the treatment and control. The lowercase letters for control and treatment in each column, and uppercase letters for % survival, which connected by the different letters are significantly different ($p < 0.05$).

| Time (h) | *Salmonella* populations (CFU/ml) | | *Salmonella* survival (%) |
|---|---|---|---|
| | Control | With CFCS | |
| 0 | $1.56 \times 10^6 \pm 1.25 \times 10^5$ [a] | $2.07 \times 10^6 \pm 1.10 \times 10^6$ [a] | >100[A] |
| 2 | $2.51 \times 10^6 \pm 9.56 \times 10^5$ [a] | $5.21 \times 10^5 \pm 2.72 \times 10^4$ [b,*] | $20.74 \pm 1.08$[B] |
| 4 | $8.95 \times 10^7 \pm 5.82 \times 10^6$ [b] | $2.90 \times 10^5 \pm 7.70 \times 10^4$ [b,*] | $0.32 \pm 0.09$[B] |
| 8 | $1.58 \times 10^8 \pm 3.37 \times 10^7$ [c] | $4.37 \times 10^4 \pm 1.04 \times 10^4$ [b,*] | $0.03 \pm 0.01$[B] |
| 12 | $3.20 \times 10^8 \pm 2.25 \times 10^7$ [d] | $2.57 \times 10^4 \pm 7.37 \times 10^3$ [b,*] | $0.01 \pm 0.00$[B] |
| 16 | $3.39 \times 10^8 \pm 2.75 \times 10^7$ [d] | $6.63 \times 10^3 \pm 1.40 \times 10^3$ [b,*] | $0.002 \pm 0.00$[B] |
| 20 | $3.56 \times 10^8 \pm 3.10 \times 10^7$ [d] | $3.47 \times 10^3 \pm 6.66 \times 10^2$ [b,*] | $0.001 \pm 0.00$[B] |
| 24 | $3.12 \times 10^8 \pm 1.76 \times 10^7$ [d] | $3.83 \times 10^3 \pm 1.63 \times 10^3$ [b,*] | $0.001 \pm 0.00$[B] |

**Table 5 Swarming zone of *Salmonella* Typhi in different concentration of WM36–CFCS.** All values represented as mean ± standard deviation ($n = 3$) and those connected by the same letters are not significant differences ($p < 0.05$).

| Concentration (%) | Swarming motility zone (mm) |
|---|---|
| Control | $45.00 \pm 5.00$[b] |
| 10 | $45.67 \pm 2.31$[b] |
| 20 | $0.00 \pm 0.00$[a] |
| 40 or MIC | $0.00 \pm 0.00$[a] |

## Influence of CFCS on *Salmonella* biofilm formation

The results of the stereomicroscopic analysis for a positive control slide showed a well-developed biofilm maturation as shown in Figs. 2A–2C, these sub-MICs (10–20 % (v/v)) showed no loose biofilm architecture. Only MIC of WM36 was clearly the most effective concentration for biofilm inhibition of *S*. Typhi indicators (Fig. 2D), compared to the negative control slide as shown in Fig. 2E.

## Influence of CFCS on virulence gene expression

In this study, virulence gene expressions, including genes responsible for T3SS, effector proteins, and QS-related genes were investigated. The different inhibitory concentrations of CFCS directly affected the significant level of virulence gene expressions with different patterns as shown in Table 6. Hence, the down-regulation of these genes could reduce virulence behaviors of *Salmonella*.

Examined genes that are responsible for T3SS and apparatus include *hilD*, *sopB*, *SopE2*, *sipA*, *sipC*, *invF* and *sptP*. When reacted to various doses of WM36–CFCS, it was found that all tested doses did the down-regulation with significant difference ($p < 0.05$) when compared with the control, except only *hilA* showed a slight down-regulation as no significant difference ($p \geq 0.05$). The *sdiA* gene performed to be significantly

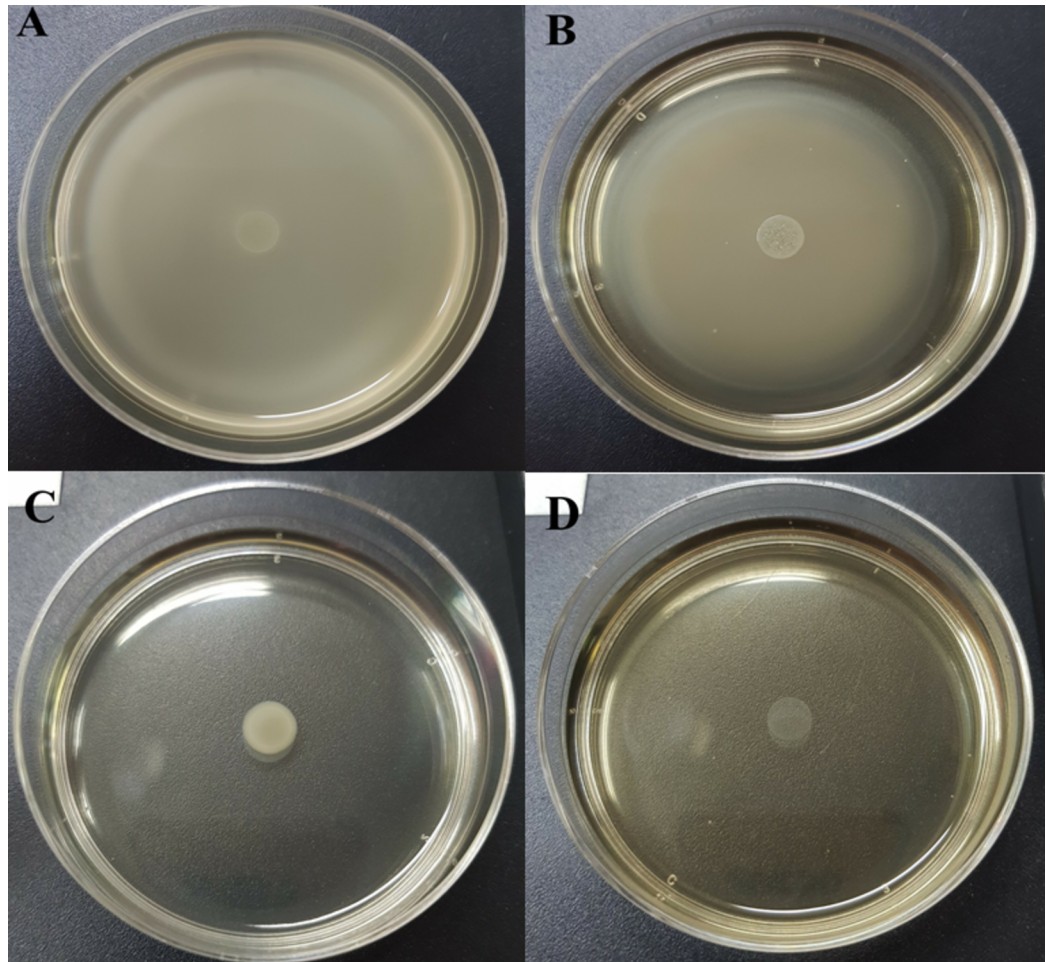

**Figure 1 Swarming motility of *S*. Typhi in the different concentration of WM36 cell-free culture supernatant.** (A) control, (B) 10% (v/v), (C) 20% (v/v) and (D) MIC (40% v/v) of CFCS. The swarming motility zone was measured (not including dropped zone) and expressed as mean ± standard deviation as previous described in Table 5.

down-regulated ($p < 0.05$) in its expression when reacted to sub-MICs and MIC of WM36–CFCS by comparing with the control, but *luxS* was only down-regulated at MIC, and there was no significant difference (Table 6). The above results indicate that each SPI-1 and effector proteins, and QS-related genes may decrease in their expressions after exposure with CFCS, the expression of these genes could be further reduced if reacted to higher concentrations of CFCS.

## DISCUSSION

The anti-salmonella activity of WM36–CFCS may be due to various antimicrobial compounds, which are secreted into culture supernatants, such as organic acids, $H_2O_2$, bacteriocins, and possibly others. These compounds are usually the primary and secondary metabolites and act as natural preservatives against spoilage microorganisms in fermented foods (*Castellano et al., 2017*). *W. confusa* is classified as heterofermentative LAB, which produces lactic acid, acetic acid, and $CO_2$ as the end metabolites after

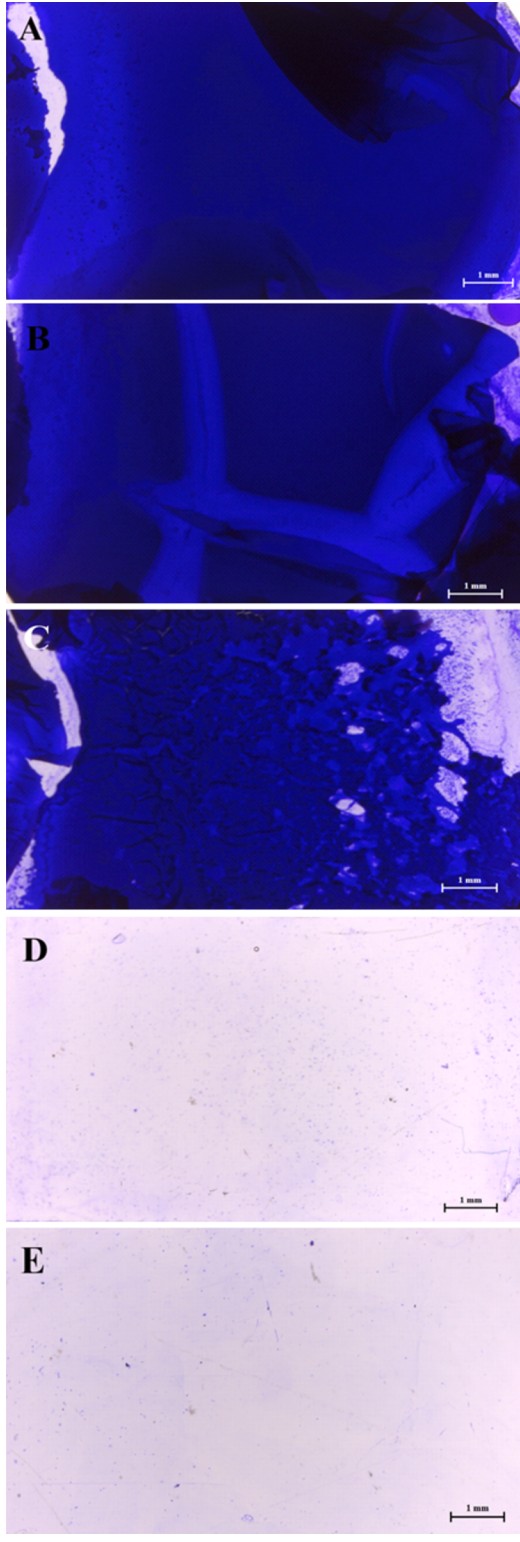

**Figure 2 Intensity of biofilm formation attached crystal violet under microscopic visualization (10× magnification) of anti-biofilm activity of *Weissella confusa* WM36 cell-free culture supernatant against *S.* Typhi.** (A) Positive control, (B) 10% (v/v), (C) 20% (v/v), (D) MIC (40% v/v) of CFCS and (E) negative control.               

**Table 6 Fold changes in virulence gene expression in *Salmonella* Typhi in the presence of different inhibitory concentration of cell-free culture supernatant from *Weissella confusa* WM36.** Values were expressed as mean ± standard deviation of duplicate. The fold-change of virulence gene was analyzed using the comparative Ct method. The values derived from $2^{-\Delta\Delta Ct}$ represent fold changes of samples in the abundance relative to the control sample (reference sample). The control sample has the $2^{-\Delta\Delta Ct}$ value of one. The fold changes of each gene expression were used to compare between treatments and control. If others showed <1 indicates the level of gene expression is decreased or if >1 indicates that the level of gene expression is increased. All superscripts represented in the same gene, which indicated significant difference ($p < 0.05$) between treatments and control.

| Genes | Fold change | | | |
|-------|-------------|------|------|-----------|
|       | Control | 10% | 20% | MIC (40%) |
| *hilA* | 1[a] | 0.68 ± 0.05[a] | 0.65 ± 0.16[a] | 0.69 ± 0.21[a] |
| *hilD* | 1[a] | 0.69 ± 0.01[b] | 0.59 ± 0.07[b] | 0.40 ± 0.01[c] |
| *sopB* | 1[a] | 0.83 ± 0.05[b] | 0.28 ± 0.03[c] | 0.12 ± 0.01[d] |
| *sopE2* | 1[a] | 0.51 ± 0.02[b] | 0.03 ± 0.01[c] | 0.02 ± 0.002[c] |
| *sipA* | 1[a] | 0.59 ± 0.00[b] | 0.29 ± 0.08[c] | 0.17 ± 0.05[c] |
| *sipC* | 1[a] | 0.81 ± 0.02[b] | 0.83 ± 0.03[b] | 0.17 ± 0.02[c] |
| *invF* | 1[a] | 0.09 ± 0.01[b] | 0.04 ± 0.002[c] | 0.02 ± 0.00[c] |
| *sptP* | 1[a] | 0.82 ± 0.01[b] | 0.62 ± 0.03[c] | 0.13 ± 0.02[d] |
| *sdiA* | 1[a] | 0.79 ± 0.34[b] | 0.73 ± 0.01[b] | 0.15 ± 0.01[c] |
| *luxS* | 1[a,b] | 1.18 ± 0.19[b] | 1.49 ± 0.16[b] | 0.60 ± 0.04[a] |

consuming sugars because of its biochemically metabolic pathways (*Björkroth, Dicks & Endo, 2014*).

Our CFCS showed lower inhibitory activity against *Salmonella* than other *Weissella* isolates, *W. confusa* Cys2-2 gave the highest inhibitory zone up to 17.66 mm and 14.00 mm against *S. enterica* UTNSm2 and *S.* Abaetetuba ATCC 35640, respectively. The inhibitory action is due to the presence of antimicrobial peptides, along with organic acids and inhibitory activity, which was also greater when using the agar well diffusion method (*Tenea & Lara, 2019*). From the obtained results, it is possible that WM36–CFCS had limited solubility in the agar medium due to the different methodological approach, and it may also depend on the pathogenic virulence of the target organism. These factors made our CFCS show lower activity than reported above. This led us to explore the antimicrobial metabolites produced by *W. confusa* WM36.

To characterize the antimicrobial substances presented in the CFCS, acid-neutralizing CFCS showed on inhibitory activity when compared to other treated CFCS. This indicated that organic acids are the nature of antimicrobial substance presented in WM36–CFCS. This concurs with previous studies by several authors, where organic acids in LAB, including WM36–CFCS, may exert antimicrobial activity vs. various serotypes of *Salmonella* growth (*Beier et al., 2017*; *Koyuncu et al., 2013*; *Mani-Lopez, García & López-Malo, 2012*; *Van Immerseel et al., 2006*).

*W. confusa* WM36 produced lactic acid and acetic acid up to 266 mm (2.6 %w/v) and 261 mm (1.6%w/v), respectively (Table 2). These findings indicate that this strain produced both organic acid higher than other *W. confusa* isolates when compared with previous works (*Akanji & Alake, 2016*; *Baek et al., 2012*; *Mukisa et al., 2017*). The amount

of lactic and acetic acid produced from *W. confusa* was measured in MRS broth and under the experiment performed in this work and was quite similar to the report of *Akanji & Alake (2016)* with the same culture condition that *W. confusa* FS027 from cabbage producing 20,480 µg/ml (2.04% w/v) and 17,184 µg/ml (1.72% w/v) of lactic and acetic acid, respectively. In contrast, the lactic and acetic acid were measured at approximately 2 g/kg (0.2% w/w) and 0.1 g/kg (0.01% w/w), respectively in *W. confusa* cultured in some cereals (*Mukisa et al., 2017*). *W. confusa* D2-96 produced lactic acid and acetic acid from rice cake up to 60.3 mm (0.54% w/v) and 41.6 mm (0.25% w/v) and exerts antifungal activity against spoilage fungi (*Baek et al., 2012*). In addition to lactic acid and acetic acid, other anti-salmonella might be produced by the WM36 strain as previously mentioned. However, this step can conclude that organic acids produced from WM36, have an important role in inhibiting growth. In order to evaluate whether organic acids in WM36–CFCS are responsible for the inhibitory activity against *Salmonella* indicator.

Beside lactic acid and acetic acid, 2,4 DTBP is also an important metabolite of WM36–CFCS, showing anti-growth activity against *S.* Typhi. 2,4 DTBP is a volatile organic acid compound, is grouped as a member of the phenol class, and plays the main role in antimicrobial and antioxidant activities (*Varsha et al., 2015*). 2,4 DTBP was found to be a microbial metabolite such as *Streptomyces* sp., *S. mutabilis*, *Bacillus licheniformis*, *B. subtilis*, *Pseudomonas monteilii*, and *Lactococcus* sp. (*Varsha et al., 2015*; *Dharni et al., 2014*; *Chawawisit et al., 2015*; *Belghit et al., 2016*; *Viszwapriya et al., 2016*; *Aissaoui et al., 2018*). This compound shows the bactericidal agent to inhibit the growth of pathogenic bacteria, such as methicillin-resistant *S. aureus*, group A streptococci and *P. aeruginosa*. 2,4 DTBP has fungicidal activity against *Aspergillus niger*, *A. carbonarius*, *Fusarium oxysporum*, *Penicillium chrysogenum*, *Candida albicans* and other pathogenic fungi (*Varsha et al., 2015*; *Dharni et al., 2014*; *Chawawisit et al., 2015*; *Belghit et al., 2016*; *Viszwapriya et al., 2016*; *Aissaoui et al., 2018*).

Additionally, 2,4 DTBP has also been recognized as an anti-biofilm agent to reduce the biofilm formation of *Serratia marcescens* and *Streptococcus pyogenes* (*Viszwapriya et al., 2016*). However, 2,4 DTBP has a limitation as it is soluble in water, with an optimum solubility of 35 mg/l at 25 °C (*National Center for Biotechnology Information, 2019*). It is a possible reason why our CFCS had a limited solubility in the agar well diffusion test and gave the lower anti-salmonella activity by this method.

The mode of action of phenols and phenolic derivatives on microorganisms are to penetrate through cell membranes and inactivate the intracellular enzymes by forming an unstable complex, to inhibit permeases that cause protein denaturation and cell membrane lysis leading to the release of nucleic acid (*McDonnell & Russell, 1999*). The phenolic compounds have been reported to show a broad spectrum of biological functions, including antimicrobial, antioxidative and anti-tumor activity. Furthermore, the compound is stable and tolerates any stress conditions. It is usually used as a disinfectant. To the best of our knowledge, this is the first report to find that *W. confusa* can produce 2,4 DTBP, and this compound may act together with organic acids to antagonize the growth of *S.* Typhi.

The results obtained from the effect of CFCS on *Salmonella* growth and survival reveal that WM36–CFCS has the potential for controlling *S.* Typhi so it would be possible to use an alternative therapy to control *Salmonella*. These results also show the reduction of *Salmonella* growth and population closely correlates with the metabolites and the inhibitory concentration of WM36–CFCS (Tables 2–4). This confirms that organic acids (lactic acid and acetic acid) including 2,4 DTBP are the main metabolites to act as anti-salmonella activity of *W. confusa* WM36.

In order to investigate the effects of different concentrations of WM36–CFCS on *Salmonella* motility, the swarming motility zone of *Salmonella* indicator depended on the concentration of CFCS (Fig. 1). The CFCS concentrations at 20% (v/v) and MIC completely reduced *Salmonella* motility and are more effective than 10%. However, WM36–CFCS was less effective than the previous reports based on the concentration used the tested volume (*Muyyarikkandy & Amalaradjou, 2017*). Only 7.5% (v/v) of CFCS from *Lactobacillus delbreuckii* subspecies *bulgaricus* and *L. paracasei* reduced *S.* Typhimurium motility up to ∼25% and ∼15%. Similarly, the use of *L. rhamnosus* CFCS with a concentration at 7.5% (v/v) resulted in a reduction of three *Salmonella* serovars (Enteriditis, Typhimurium, and Heidelberg), with motility increased to 40%, 30% and 20%, respectively (*Muyyarikkandy & Amalaradjou, 2017*). However, the pathogenic virulence strains should be considered as well.

Biofilms are one of the virulence factors involved in the pathogenesis of *S.* Typhi. This virulence aids *Salmonella* to survive within stress environments and to resist antibiotics by increasing the efflux pump and enhancing exopolysaccharide production (*Peng, 2016*). Several LAB and their metabolites were investigated for their ability to inhibit the biofilm formation, and to decrease the number of antibiotic-resistant, biofilm-producing pathogens. For example, *Lactobacillus rhamnosus* GG can produce lectin-like molecules, namely Llp1 and Llp2 to interfere with the biofilm formation of *S.* Typhimurium. These proteins are involved in the adhesion capacity of *L. rhamnosus* GG to gastrointestinal epithelium (*Petrova et al., 2016*). Moreover, some organic acids are an efficient bioactive compound which interfere with the biofilm formation of *S.* Typhimurium; they yield the maximum biofilm inhibition ranging from 13% to 39% and also play the main role to inhibit the exopolysaccharide production (the main composition of biofilm) (*Amrutha, Sundar & Shetty, 2017*). In this study, WM36–CFCS at MIC showed a reduction of biofilm formation. The findings strongly suggest that our *Weissella* and its metabolites such as organic acid and 2,4 DTBP may be a new strategy to fight biofilm-producing *Salmonella* and other pathogens.

The invasion of host epithelial cells is mediated by T3SS, which require a set of genes within SPI-1. The examined genes including *hilA*, *hilD*, *sipA*, *sipC*, *invF* and *sptP* are located in SPI-1 and play the main role in cell invasion of *Salmonella*. The expressions of these genes have required the regulation of different regulators, while *hilA* gene plays the main regulator (*Valdez, Ferreira & Finlay, 2019*). *hilA* gene appears to directly activate the *invF* signaling cascade resulting in the expression of effector genes, *sopB*, and *sopE2*, which are located in SPI-5. All the above genes are manipulated and work together as a

signaling network to form syringe machinery and deliver effector proteins to destruct host cell cytoskeleton (*Marcus et al., 2000*).

Several LAB and their metabolites can reduce the expression of SPI-1 and related genes. Our work is consistent with cited reports, which have been reported by other authors. For example, the neutralized cell-free supernatants of four probiotics including *L. acidophilus*, *L. plantarum*, *B. bifidum* and *B. infantis* down-regulate the reporter gene expression of *hilA* of *S.* Typhimurium when CFCS was added at 50% (v/v) (*Bayoumi & Griffiths, 2010*). *Lactobacillus zeae* culture fluid also down-regulate the expression of virulence genes responsible for T3SS and effector proteins (*hilA*, *hilD*, *sopB*, *sopE2*, *sipA* and *sptP*) of *S.* Typhimurium in vitro (*Yang et al., 2014*). *L. delbreuckii* NRRL B548, *L. rhamnosus* NRRL B442 and *L. paracasei* DUP-13076 culture supernatants at 7.5% (v/v) concentrations also down-regulated the expression of *hilA*, *hilD*, *sipA* and *sopB* of *S.* Enteritidis, *S.* Heidelberg and *S.* Typhimurium (*Muyyarikkandy & Amalaradjou, 2017*). While further research is encouraged, this is the first report that *W. confusa* can reduce the expression of SPI-1 and effector genes of *S.* Typhi. This suggests that WM36–CFCS has a greater potential to reduce the virulence of *S.* Typhi.

Numerous bacteria regulate gene expression in the response of changes in their population density through the process called the QS signaling system. QS systems are associated with bacterial pathogenesis (*Rutherford & Bassler, 2012*). Bacteria are able to produce, release and detect small signaling molecules termed as autoinducer (AI), which mediate bacterial communications. *S. enterica* has at least three types of AI signaling systems; the AI-1/LuxIR system, the AI-2/LuxS system, and the AI-3 system (*Parker & Sperandio, 2009*; *Walters & Sperandio, 2006*).

The AI-1/LuxIR system is usually found in Gram-negative bacteria; these bacteria consist of LuxI synthase, which plays a role in producing acyl-homoserine lactones (AHLs) as AI-1 molecules (*Miller & Bassler, 2001*). AHL is recognized by the LuxR receptor, which is encoded by the *luxR* gene, while the LuxI synthase is encoded by the *luxI* gene. After binding, the signaling cascades are activated and subsequently, several genes are expressed (*Parker & Sperandio, 2009*; *Sifri, 2008*). *Salmonella* does not produce AHL due to the lack of the *luxI* gene, but can detect AHLs by SdiA, the LuxR homology (*Ahmer et al., 1998*; *Dyszel et al., 2010*; *Sperandio, 2010*). AI-1/LuxIR of *Salmonella* is necessary to regulate the expression of several genes in virulence plasmid, especially the *rck* gene, which aids *Salmonella* to resist the host immune defense (*Ahmer et al., 1998*; *Soares & Ahmer, 2011*).

The AI-2/LuxS system has also been found to be a universal QS system in both Gram-negative and Gram-positive bacteria, and mediates inter-species specific communication. AI-2 molecules are produced by the LuxS enzymes, which are encoded by the *luxS* gene, then released into environments through membrane transporter proteins. Extracellular AI-2 can bind the autoinducer binding protein LsrB on the bacterial cell surface, and are transported into the cell cytoplasm via the Lsr apparatus, then the signal cascades are activated (*Federle, 2009*; *Pereira, Thompson & Xavier, 2013*; *Pui et al., 2011*). The AI-2/LuxS system is important for *Salmonella* invasion; this system is activated

by the expression of SPI-1 proteins which are responsible for T3SS formation (*Choi, Shin & Ryu, 2007*).

In the case of the AI-3 system, signaling molecules are accomplished by a two-component system, comprised of histidine sensor kinase QseC and the response regulator QseB. The AI-3 production and structure are still unclear. QseC undergoes autophosphorylation and transfers a phosphate group to QseB, which activates a gene responsible for flagella biosynthesis and motility (*Parker & Sperandio, 2009*). In addition, the QseB regulator of the host's normal flora is also recognized for epinephrine, norepinephrine and neurotransmitter. This system induces the SPI-2 gene expression to support *Salmonella* survival in macrophage, as well as facilitates the expression of gene encoded on SPI-1 and SPI-3 (*Bearson & Bearson, 2008*; *Gart et al., 2016*; *Moreira, Weinshenker & Sperandio, 2010*).

In this present work, the *sdiA* and *luxS* only, were chosen for the QS genes by interfering with the expression of these genes with CFCS. They have clear known signaling pathways, and are associated with the critical induction of SPI genes. The down-regulation of these genes might reduce the expression of signaling cascades that lead to inhibit mechanisms of *Salmonella* pathogenesis.

The examined virulence genes showed variety in their expressions; the expressions of these genes may down-regulate depending on the concentrations of CFCS. The highest tested concentration gave a highly reduced expression compared to lower concentrations. Hence, CFCS at MIC or >MIC values are the most effective inhibitory concentrations to control the levels of virulence gene expressions.

## CONCLUSIONS

This study revealed the anti-*S.* Typhi activities of *W. confusa* WM36. This LAB isolate was able to produce the antibacterial growth, survival and anti-virulence substances, which were characterized as major compounds of organic acids (mainly lactic acid and acetic acid) and 2,4 DTBP. WM36–CFCS at MIC reduced the number of *S.* Typhi populations of more than 99.99%. The sub-MIC and MIC of WM36–CFCS showed the inhibitory patterns on motility, biofilm formation, and virulence gene expression of *Salmonella* indicator, the reduction of these virulence behaviors indicating a significant decrease in virulence ability on bacterial pathogenesis of typhoid/MDR-*Salmonella*. Therefore, *W. confusa* WM36 and/or its metabolites could be potentially used as alternative therapies for controlling the *Salmonella* infection and/or use alongside with a usual treatment. However, the efficacy and safety evaluation of *W. confusa* WM36 and efficacy of each purify metabolite in vitro and in vivo have no adequate supporting information. These findings will be corroborated in the future research.

## ACKNOWLEDGEMENTS

We thank all the members in our academic group for helping us complete the experiments. The authors also acknowledge the Faculty of Pharmacy, Chiang Mai University, Chiang Mai, Thailand for kind assistance in allowing us to conduct the research work.

### Funding
This study was supported by 50th Anniversary Chiang Mai University-Ph.D. scholarship, grant number: Ph.D.010/2556 and National Research Council of Thailand (NRCT), grant number: 2560A10402021. The funders had no role in study design, data collection and analysis, decision to publish, or preparation of the manuscript.

### Grant Disclosures
The following grant information was disclosed by the authors:
Chiang Mai University-Ph.D. scholarship: Ph.D.010/2556.
National Research Council of Thailand (NRCT): 2560A10402021.

### Competing Interests
The authors declare that they have no competing interests.

### Author Contributions
- Wattana Pelyuntha conceived and designed the experiments, performed the experiments, analyzed the data, prepared figures and/or tables, authored or reviewed drafts of the paper, and approved the final draft.
- Chaiyavat Chaiyasut conceived and designed the experiments, authored or reviewed drafts of the paper, and approved the final draft.
- Duangporn Kantachote conceived and designed the experiments, authored or reviewed drafts of the paper, and approved the final draft.
- Sasithorn Sirilun conceived and designed the experiments, performed the experiments, authored or reviewed drafts of the paper, and approved the final draft.

### Data Availability
The raw data are available in Tables S1–S3, Dataset S1 and Figs. S1 and S2.
*Weissella confusa* WM36 is permanently deposited in the Thailand Bioresource Research Center (TBRC), Pathum Thani, Thailand: TBRC11086.

### Supplemental Information
Supplemental information for this article can be found online at http://dx.doi.org/10.7717/peerj.8410#supplemental-information.

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
