# Peer review of "Organic acids and 2,4-Di-tert-butylphenol: major compounds of Weissella confusa WM36 cell-free supernatant against growth, survival and virulence of Salmonella Typhi"

_PeerJ, doi:10.7717/peerj.8410_

## Round 0.1 · original submission · Minor Revisions

Please follow the recommendations provided by both reviewers.

·

Basic reporting

Pelyuntha et al, evaluated antimicrobial proprieties of cell-free culture supernatant from Weissella confuse a lactic bacteria in where other groups have been found the same proprieties, however, authors shown that this strain is able to produce high levels of antimicrobial molecules during its growth, which may be relevant in the development of probiotics. Nevertheless, there are some data that should be reviewed:

Experimental design

No comment

Validity of the findings

1.-As conclusion of the of metabolites in CFCS by GC-MS analysis, the authors argue that the main component is “2,4 DTBP, with the highest % peak area of 59.24”, however these results do not shown peaks from lactic acid and acetic acid, that should be in a high levels. What causes that these substances cannot be detected by GC-MS? Both organic acids were not reported in the table 3.
2.-The authors propose that organic acids are the main responsible for antimicrobial activity. Because addition of alkali can increase pH and neutralize the acids, however when pH is modified, also many chemical groups from other molecules (not only acids) could lose its positive charges or even modify its structure. If one of these molecules were the main antibacterial component, it could lose its antibacterial activity? What additional evidence could you provide in order to demonstrate that organic acids are in fact the responsible for this activity?
3.-I recommend to review results about HPLC and quantification of acetic and lactic acids, because 261mM and 266 mM from theses acids, would represent approximately 1.6% and 2.4% (w/v) respectively. That is a minimum of 4% of these acids in the supernatant. It is possible that Weissella confuse can produce those levels of organic acid in medium MRS under the experimental incubation conditions performed in this work? Could you explain and show some evidences (may be articles agree with your data) that support your results.

Reviewer 2 ·

Basic reporting

In Pelyuntha et al, the authors examine the effect of acid organics and 2,4,Di-tert-butylphenol of Weissella confusa WM36 under general pathogenesis of Salmonella Typhi. They found that adding the cell-free culture supernatant of W. confusa leads to decreased growth, biofilm formation, motility, and expression of virulence genes (mainly those that code for the type 3 secretion system effector proteins and quorum sensing. Besides, they found the main antibacterial compounds present in the supernatant of W. confusa, which they suggest could be good candidates molecules against typhoid fever.

Experimental design

The experiments are rigorous and the data presented clearly show an effect of the cell-free culture supernatant of W. confusa on growth, biofilm formation, motility and gene expression of some virulence factors of Salmonella Typhi.

Validity of the findings

No comment

Additional comments

The study is straight-forward and written in a clear, logical manner, and the data appear to be rigorous. My major concern is that the data cannot be extrapolated to infection, as adding the cell-free culture supernatant of Weissella confusa to broth cultures is very different than the bacteria interacting with organic acids and 2,4 DTBP. However, this paper could be promising for further studies, which analyze more the action of W. confusa metabolites against typhoid fever. Overall, authors used appropriate methods and report results in accessible form. Some of their interpretations could be modified/expanded.

---

## Round 0.2 · accepted · Accept

After considering your revised version, it is my decision to accept this version for publication in PeerJ, congratulations!

Reviewer 2 ·

Basic reporting

No comment

Experimental design

No comment

Validity of the findings

No comment

Additional comments

The new manuscript fulfill all recommendations and suggestions of both reviewers, so I consider that now this paper is suitable for publication.

Reviewer 3 ·

Basic reporting

No comment

Experimental design

No comment

Validity of the findings

No comment

Additional comments

The results describe how the hypothesis raised during the project was fulfilled. It is a very complete and interesting manuscript.